

# Early detection of *Cercospora beticola* and powdery mildew diseases in sugar beet using uncrewed aerial vehicle-based remote sensing and machine learning

Koç Mehmet Tuğrul[1], Rıza Kaya[2], Kemal Özkan[3], Merve Ceyhan[3], Uğur Gürel[3] and Fatih Yavuz Fidantemiz[4]

[1] Faculty of Agriculture, Osmangazi University, Eskişehir, Turkey
[2] Plant Protection Department, Turkish Sugar Factories Corporation Sugar Institute, Etimesgut, Ankara, Turkey
[3] Faculty of Engineering and Architecture, Department of Computer Engineering, Osmangazi University, Eskişehir, Turkey
[4] Transitional Zone Agricultural Research Institute Directorate, Agricultural Irrigation and Land Reclamation Unit, Republic of Türkiye Ministry of Agriculture and Forestry, Eskişehir, Turkey

Corresponding author
Koç Mehmet Tuğrul,
kmtugrul@gmail.com

## ABSTRACT

**Background:** Agricultural production is crucial for nutrition, but it frequently faces challenges such as decreased yield, quality, and overall output due to the adverse effects of diseases and pests. Remote sensing technologies have emerged as valuable tools for diagnosing and monitoring these issues. They offer significant advantages over traditional methods, which are often time-consuming and limited in sampling. High-resolution images from drones and satellites provide fast and accurate solutions for detecting and diagnosing crops' health and identifying pests and diseases affecting them.

**Methods:** The research focused on the early detection of *Cercospora* leaf spot (*Cercospora beticola Sacc.*) and powdery mildew (*Erysiphe betae (Vaňha) Weltzien*), which cause significant economic losses in sugar beet before visible symptoms emerge. The study was accomplished by capturing images of uncrewed aerial vehicle (UAV) in field conditions. To effectively evaluate different detection methods in agricultural contexts, the study targeted two key areas: (1) monitoring *Cercospora* in fields without pesticide application, utilizing the Metos climate station early warning system alongside UAV-based image analysis, and (2) monitoring powdery mildew, which involved visual disease detection and targeted spraying based on UAV image processing. Trial plots were established for this purpose, with six replications for each method.

**Results:** UAV-based images show that Normalized Difference Vegetation Index values in leaves decreased before disease onset. This change is an important warning sign for the emergence of the disease. Additionally, the study demonstrated that early detection of diseases is possible using K-nearest neighbors and logistic regression algorithms, exhibiting high discrimination and predictive accuracy.

## INTRODUCTION

Sugar beet (*Beta vulgaris*) is an important agricultural product that plays a significant role in essential nutrition. It also has considerable potential for bioenergy, industrial raw materials, and food production. However, sugar beet cultivation often faces significant yield and quality losses due to diseases, a fundamental challenge in sugar production. Two common diseases that inflict severe damage to crops like sugar beet and corn are *Cercospora* leaf spot (CLS) and powdery mildew (PM). It is crucial to effectively control these diseases and pests to increase agricultural production in a sustainable and high-quality manner. Traditional diagnostic methods can be ineffective, especially when the symptoms of the disease are mild or barely visible, leading to difficulties in regional diagnosis. One of the most widely used vegetation indices in remote sensing for monitoring plant health is the Normalized Difference Vegetation Index (NDVI). The significant relationship between NDVI and green biomass allows for high-accuracy monitoring of plant development in remote sensing studies (*Jin & Eklundh, 2014*). In this context, the importance of remote sensing and advanced technologies in agricultural management is growing.

Drone remote sensing offers several advantages, including high spatial resolution, efficiency, operational flexibility, reliability, low cost, and extensive coverage. These benefits enable early detection of diseases in crops. The concept of drone disease detection involves automatic image capture, feature extraction, and the application of machine learning or deep learning algorithms for disease diagnosis. Drones equipped with RGB, multispectral, hyperspectral, and thermal infrared imaging sensors and data analysis algorithms are essential for minimizing potential losses and improving efficiency by identifying early-stage damage factors that could impact agricultural products (*Abbas et al., 2023*). Traditional disease detection methods typically rely on identifying visible symptoms and pathogens, which are assessed by experts whose evaluations may vary based on personal experience and are influenced by temporal changes (*Chen et al., 2018*). In contrast, remote sensing systems that utilize cameras and machine learning algorithms facilitate rapid and accurate disease detection in the field, enabling real-time monitoring of plant health and improved disease management (*Kuska et al., 2015*). Additionally, accessible satellites with a spatial resolution ranging from 10 to 60 m can produce and process data integrated with artificial intelligence models, providing valuable images for monitoring crop health and detecting plant stress conditions (*Segarra et al., 2020*; *ListenField, 2023*).

Plant density per unit area is the primary factor causing yield losses in sugar beet production. CLS does not develop in sugar beet when air temperatures are below 10 °C. Sporulation, germination, and infection by the disease agent, *Cercospora beticola*, occur at daytime temperatures between 25–35 °C and nighttime temperatures above 16 °C. These conditions require prolonged high relative humidity (over 90%) or free moisture on the leaves. CLS conidia form most rapidly at temperatures of 20–26 °C with 90% to 100% relative humidity, and there should be at least 8.5 h of leaf wetness. Various fungal factors, including CLS disease (*Mohamed Gouda, El-Naggar & Yassin, 2022*) and root rot,

significantly contribute to decreased number of plants (*Esh & Taghian, 2022*; *Tan et al., 2023*). Early diagnosis and effective treatment of these diseases are crucial for increasing sugar beet yield and minimizing losses.

Machine learning and deep learning architectures, when applied to the analysis of physical disease symptoms, including those of CLS, can achieve high accuracy in disease detection (*Kamilaris & Prenafeta-Boldú, 2018*; *Mohanty, Hughes & Salathé, 2016*; *Ramkumar et al., 2021*). The support vector machine (SVM) algorithm is often favored in studies because it demonstrates a prediction success rate of 65% to 90% when detecting disease agents using spectral vegetation index data (*Rumpf et al., 2010*). A significant challenge in image classification with deep learning is the requirement for a large image database to enhance prediction accuracy. Although data augmentation techniques address this issue, they do not always produce satisfactory results. Therefore, a 12% increase in prediction accuracy is achievable when training data focuses on disease lesions and spots rather than the entire leaf, allowing for identifying a greater variety of diseases (*Barbedo, 2019*).

Uncrewed aerial vehicle (UAV)-based approaches for plant disease detection and monitoring, which utilize sensors and cameras, offer fast and effective solutions by capturing high-resolution and spectral images. These tools can detect even minute changes in crop development (*Shahi et al., 2023*). A ground sampling distance (GSD) of 0.3–0.5 cm when using UAV has proven effective in monitoring crop development, particularly regarding diseases (*Yamati et al., 2022*). Four different wavelength ranges, 500–533 nm, 560–675 nm, 682–733 nm, and 927–931 nm, are commonly used to classify healthy and diseased plants (*Bauriegel et al., 2010*). Two primary methods for early disease prediction are the Continuous Change Detection and Classification (CCDC) algorithm and the NDVI trend. Both techniques analyze data from spectral bands or indices to monitor linear trends. Breakpoints are identified pixel-by-pixel, and differences in spectral segments are assessed as indicators of plant stress (*Shumway & Stoffer, 2017*; *Lasaponara, Abate & Masini, 2024*). A study that employed UAV and convolutional neural networks (CNN) to detect levels of CLS development indicated a high degree of accuracy in disease detection (*Görlich et al., 2021*).

Furthermore, evaluating image processing methods for early disease risk detection and climate data modeling enhances the success rate. Specifically, a combination of 90% relative humidity during 4 h of continuous rain, 60% relative humidity for the subsequent 9 h, daytime temperatures of 16 °C, and nighttime temperatures exceeding 10 °C creates an environment conducive to CLS development. Predicting disease occurrence based on models grounded in these conditions and implementing a spraying program can be an effective disease control strategy (*El Jarroudi et al., 2021*). In wheat affected by PM stress, spectral reflectance varies due to physiological and biochemical changes and alterations in leaf temperature caused by a decrease in chlorophyll content (*Khan et al., 2021*; *Awad et al., 2015*). Compared to traditional single-image methods, the k-nearest neighbors (KNN) approach that utilizes multitemporal images provides a more comprehensive dataset regarding disease development, facilitating high-accuracy evaluations (*Ma et al., 2018*).

This study aims to diagnose the physiological changes that lead to plant disease during the early stages by monitoring plant development using UAV, satellite, and ground-based multispectral images. This approach will allow for the detection of potential CLS and PM diseases in sugar beets before physical symptoms appear. By identifying these issues at the right time, we can reduce economic losses and minimize the environmental impact caused by unnecessary pesticide applications. Additionally, this research will provide an effective method for promoting sustainable production by decreasing crop losses due to diseases.

## MATERIAL AND METHOD

### Location

The CLS trial was established in the Karaköy area of the Bursa-Yenişehir region, with coordinates 40°16′27.09″N, 29°34′26.15″E, and 40°17′3.37″N, 29°34′47.15″E during the years 2022 and 2023, when the disease was widely observed. In the same timeframe, the PM trial was set up in the Eskişehir Center, specifically at the Karacahöyük location (39°45′27.02″N, 30°36′6.67″E) and the Karagözler location (39°46′29.25″N, 30°24′17.75″E) (Fig. 1).

### Material

#### *MS imaging*

The Phantom 4 MS drone, manufactured by DJI in Shenzhen, China, captured multispectral (MS) images. This drone has six 1/2.9″ CMOS sensor cameras, including one RGB sensor for capturing visible light images and five monochrome sensors dedicated to multispectral imaging. Each of these sensors has a resolution of 2.12 MP, with the following specifications: blue (B) at 450 ± 16 nm, green (G) at 560 ± 16 nm, red (R) at 650 ± 16 nm, edge (RE) at 730 ± 16 nm, and near-infrared (NIR) capturing images in the 840 ± 26 nm band.

Flights were consistently conducted between 10:00 and 14:00 under sunny conditions, with wind speeds not exceeding 12 ms$^{-1}$. Images were captured 15 m above ground level (AGL), resulting in a ground sampling distance (GSD) of approximately 0.4 cm. The images' front and side overlap ratios were 80% and 70%, respectively. Under these flight parameters, roughly 312 MS images were taken during each flight with the DJI P4 MS drone. The images were then radiometrically corrected using the Mapir T3-R125 calibration plate.

An orthophoto, a high-resolution aerial image, was carefully created by combining the received parcel images using the advanced Pix4D Mapper version 4.6.4 (Fig. 2). Each parcel, representing different subjects on the reflection map, was marked according to its region, and separate NDVI maps were generated using an index calculator. In this research, we utilized the sophisticated Jenks classification method for classifying the index maps (*Jenks, 2023*). This data clustering method effectively highlights the differences between classes by minimizing variance within each class while maximizing variance between classes. The classification methods produce five color classes based on pixel differences (*Pix4D, 2023*). The number of color classes can be adjusted according to the sensitivity required for the analysis. In this study, we relied on the differences between the
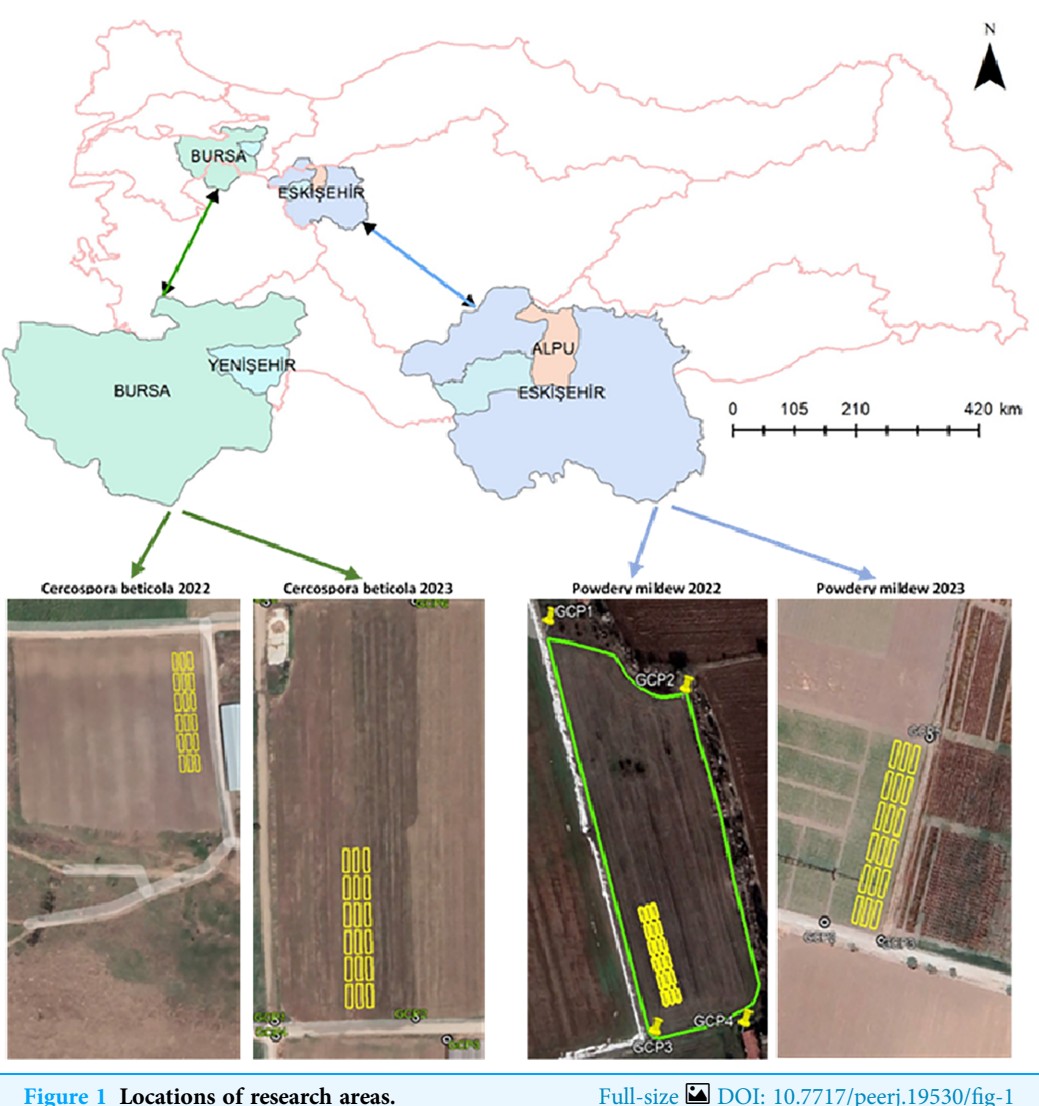

**Figure 1** **Locations of research areas.**

color classes to distinguish between varietal differences and characteristics, such as drought resistance (Fig. 2).

The changes in numerical NDVI index values obtained from the trial subjects were analyzed using Excel, and the disease development status was presented graphically. Additionally, evaluations of the 10 m resolution NDVI maps collected by Metos *via* the Sentinel 2A satellite, terrestrial NDVI imaging, and measurements obtained with the GreenSeeker 505 were also performed in Excel.

### Terrestrial NDVI measurement

Local NDVI values were measured using the NTech GreenSeeker Model 505 (GS) optical hand sensor to compare with drone-based measurements. This device operates at 660 nm (red) and 770 nm (NIR) wavelengths to calculate NDVI. It functions based on the principle of spectral reflection, calculating values based on reflections at different wavelengths (*Peňuelas et al., 1993*). For the readings, the sensor was held 80 cm above the vegetation
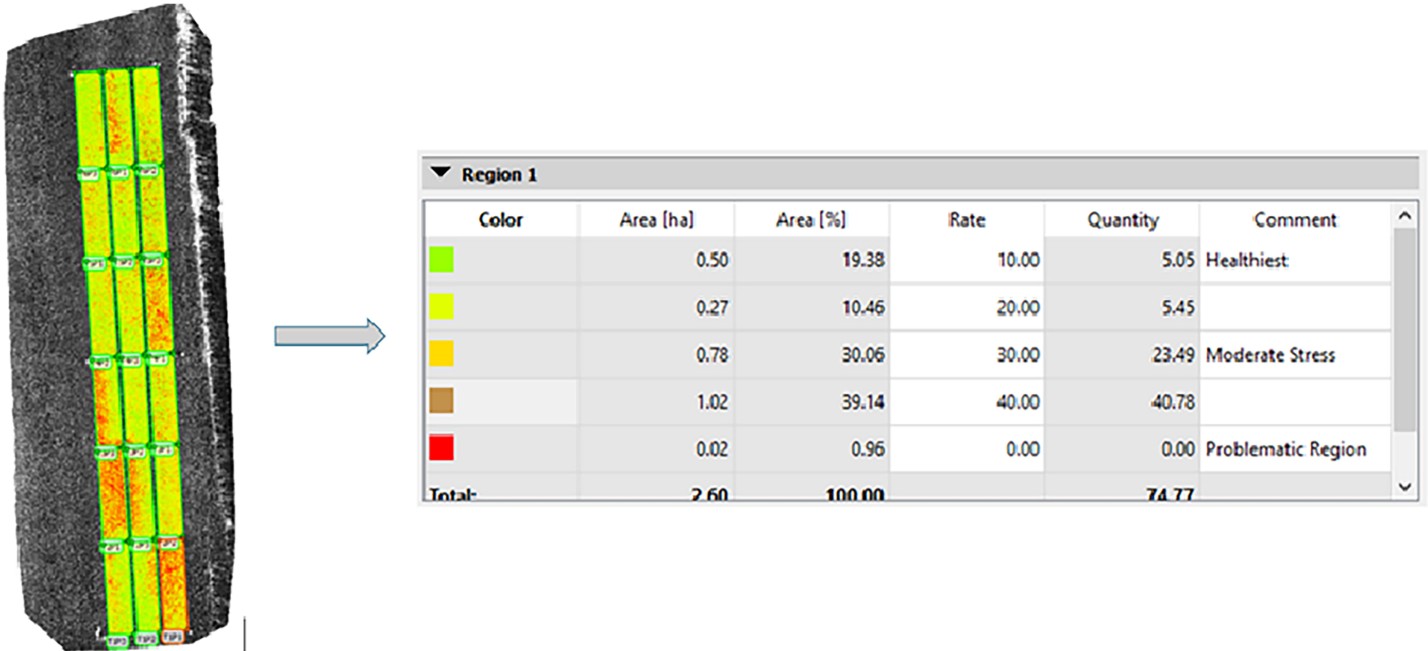

**Figure 2 Index map created by Jenks classification method.**

and moved along the length of the parcel at a constant speed of 5 km h$^{-1}$. In each trial plot, five measurements were taken in the same order.

### Climate station

The iMETOS 3.3 climate station, utilized in the second issue of the CLS trial, was employed to facilitate pesticide application based on alerts from the early warning software. This station has sensors that measure air temperature, relative humidity, rainfall, global radiation, wind speed, and leaf wetness. These measurements are essential for disease modeling and calculating evapotranspiration. The system is powered by solar energy and uses the University of Minnesota's CE software, which calculates daily infection values (*Shane & Teng, 1984*) for early warning purposes. Additionally, it incorporates risk formation software that evaluates incubation and sporulation processes (*Bleiholder & Weltzien, 1972*).

### Method

In the study, images captured in the multispectral (MS) band over time using a UAV were processed with Pix4D software to generate digital surface model (DSM) maps. An early-stage disease prediction method was developed based on the hypothesis that each break in the NDVI index values, as represented in the DSM map and which increases linearly with the growth of sugar beet, is related to climate data and the progression of the disease (*Lasaponara, Abate & Masini, 2024*).

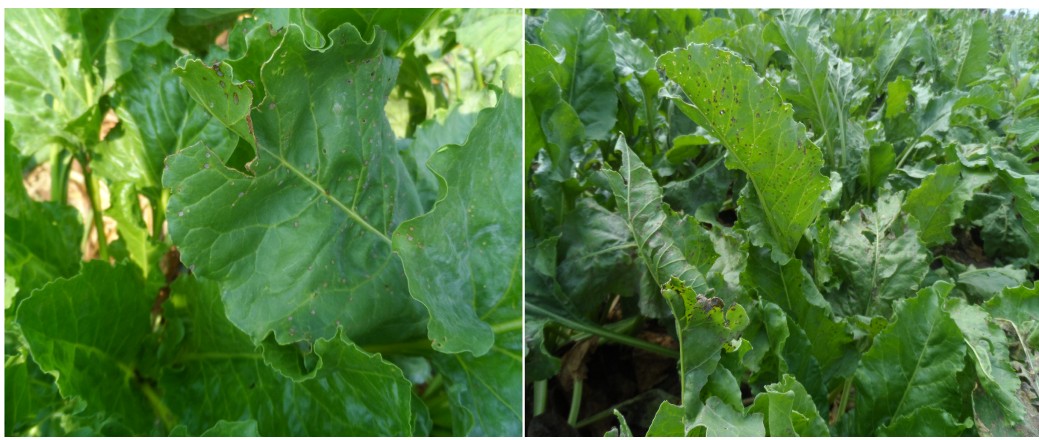

**Figure 3 CLS disease, which first occurs (left) and covers the entire leaf (right).**

### CLS trial

CLS is a common leaf disease caused by fungi of the *Cercospora* genus. This disease affects various plant species but is most frequently found on sugar beets and leafy vegetables. The fungi settle on the leaves, resulting in the formation of brown spots (Fig. 3). These spots expand over time, hindering photosynthesis, weakening plant growth, reducing yield, and leading to quality losses. To control this disease, farmers may use chemicals, cultural practices, and disease-resistant varieties (*Esh & Taghian, 2022*; *Tan et al., 2023*).

CLS typically reaches its damage threshold by mid-June, depending on the regional climate, prompting the start of spraying treatments. In trial plots, observations of CLS were conducted according to the KWS scale. This involved randomly selecting 25 plants from the middle three rows at 15–20 day intervals. The KWS scale ranges from 1 to 9, where 1 indicates the first appearance of spots on the outer leaves, 3 signifies spots on intermediate leaves (not including the middle leaves), 5 is assigned when large dead areas develop on the leaves, 7 is given when dead regions are present on at least half or more of the outermost leaves, and 9 indicates the formation of new leaves on the plants (*Yamati et al., 2022*). Spraying commenced according to the experimental protocols and continued at intervals of 15–20 days until 1 month before harvest. The treatment used was Amistar Gold SC (12.5%), which is recommended by the Ministry of Agriculture and Forestry and contains Azoxystrobin and Difenoconazole (12.5%) as the active ingredients (*BKU, 2023*). In addition to visual observations, the collected data were analyzed using the Townsend-Heuberger and Abbott formulas to calculate disease rates and fungicide efficiencies (*Wieninger & Kubadinov, 1971*) and were evaluated using the MS-Stat program.

### PM trial

PM is a widespread fungal disease affecting sugar beet plants. In severe cases, it manifests as a white coating on the leaves, which can ultimately lead to leaf death (Fig. 4). This disease weakens plant tissues and causes leaf drying, reducing yield and quality (*Ata et al.,*

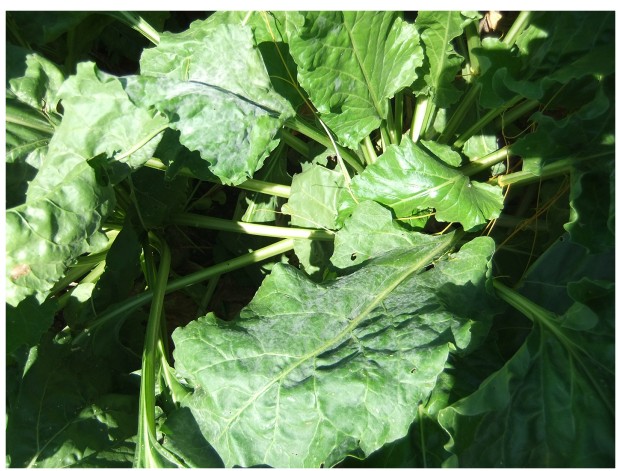

**Figure 4 PM disease covering the entire leaf.**   

*2023*). The occurrence and intensity of PM can vary significantly depending on the climate of a given year. Following the initiation of spraying in field trials, treatments were carried out every 15–20 days until 1 month before harvest. The fungicide was sulfur-based FRS Glok 80 WG, which contains 80% sulfur as its active ingredient (*BKU, 2023*). Fifteen to twenty days before harvest, PM observations were conducted on one older leaf from 25 randomly selected plants in each plot. The severity of the disease was rated on a scale from 0 to 5, where 0 indicates no disease present, 1 signifies 1–10% disease occurrence on the leaves, three represents 36–65% disease occurrence, and five means more than 91% disease occurrence (*TAGEM, 2017*).

### Field applications

Sugar beet planting in the CLS field occurred on April 4, 2022, in the first year and on March 26, 2023, in the second year. Harvesting was conducted on October 25, 2022, and October 26, 2023. To examine the differences between the research subjects in the project, three treatments were implemented: control (S1), early spraying in conjunction with multispectral imaging to diagnose disease symptoms (S2), and spraying according to the Metos climate station CLS early warning system (S3). The field trial was designed as a randomized block trial with six replications (Fig. 5). Imaging and analysis were performed on 18 plots, each measuring 2.7 × 10 m and covering an area of 27 m$^2$.

For the PM field, sugar beet planting occurred on May 3, 2022, in the first year, and on May 25, 2023, in the second year. Like the CLS trial, the trial design consisted of three treatments with six repetitions. Treatments S1 and S2 were identical to those in the CLS trial. In contrast, in treatment S3, fungicide application was made immediately upon observing the first physical symptoms, again due to the lack of an early warning system for PM disease. All other agricultural treatments were conducted similarly across the trial plots. The trial plots were harvested on November 3, 2022, in the first year, and October 21, 2023, in the second year. The Terranova (KWS) variety, widely cultivated in the region,
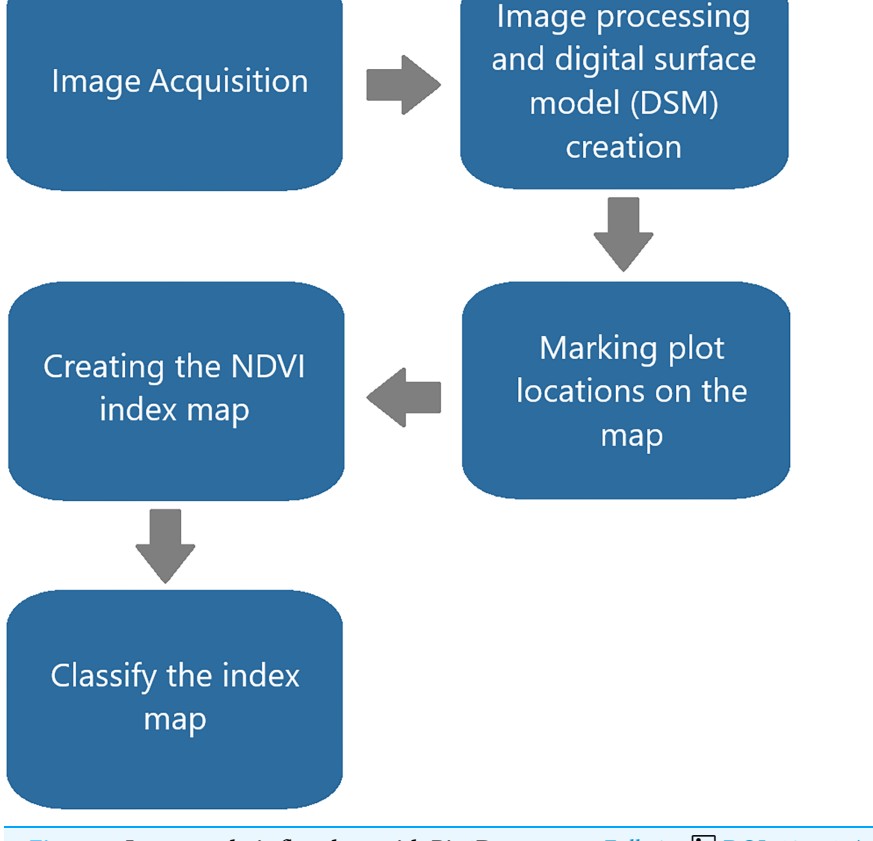

**Figure 5** **Image analysis flowchart with Pix4D.**

was used for planting in both years. Yield and quality analyses of samples from a 10 m$^2$ area were applied as in CLS.

## MACHINE LEARNING ANALYTICS

The research utilized datasets for both CLS and PM diseases. Various values, including green, yellow, orange1, orange2, and red, were considered during the MS evaluation to facilitate early detection of these diseases. Additionally, a climate dataset containing information on temperature, humidity, precipitation, wind, and leaf wetness for the specified days was incorporated. For the CLS disease, the dataset included average, maximum, and minimum temperature, humidity, precipitation, and leaf wetness data. For the PM disease, the dataset consisted of average and maximum temperature and humidity, precipitation, average wind, and maximum wind data. The initial dataset was collected 75–80 days after planting the sugar beet. Data concerning the occurrences of CLS and PM diseases were recorded (Table 1). In the tables 'Other' column, the 'Disinfestation' information indicates whether any pesticide was applied to the agricultural land before the appearance of the disease. The terms 'Cercospora' and 'Kulleme' denote the presence of the respective diseases in the measured area. The 'Date' column provides details about when the measurements were taken. The study analyzed 19,140 data points for CLS disease,

**Table 1 Dataset contents used in the study.**

| | | CLS dataset content | | PM dataset content | | |
|---|---|---|---|---|---|---|
| MS | Temperature | Dampness | Other | Dampness | Wind | Other |
| Dampness | Temp_Mean | Damp_Mean | Spraying | Damp_Mean | Wind_Mean | Spraying |
| Damp_Mean | Temp _Max | Damp_Max | CLS | Damp_Max | Wind_Max | PM |
| Damp_Max | Temp _Min | Damp_Min | Date | Damp_Min | | Date |
| Damp_Min | | Precipitation | | Precipitation | | |
| Precipitation | | Leaf_Wetness | | | | |

comprised of 18,900 training samples and 240 testing samples, and 16,440 data points for PM disease, consisting of 16,200 training samples and 240 testing samples. Various machine learning algorithms were employed for the analysis, including logistic regression (LR) (*Sperandei, 2014*), KNN (*Peterson, 2009*), Gaussian naive Bayes (GNB) (*Ontivero-Ortega et al., 2017*), support vector machines (SVM) (*Hearst et al., 1998*), and decision tree classifier (DTC) (*Quinlan, 1996*). The analyses performed binary classification on the CLS and PM disease datasets based on the disease presence or absence information.

# RESULTS

## Climate data

The average air temperature in the CLS field was relatively stable, ranging from 15–25 °C in 2022 to 20–27 °C in 2023. High temperatures were recorded in July and August, consistent with regional norms. Relative humidity was slightly lower in the first year, averaging between 58.20% and 92.66%, compared to 63.7% to 99.89% in the second year. The CLS region experiences low precipitation; between June and September, 418 mm of rain fell in the first year and 496 mm in the second year. High relative humidity is typically linked to rainy days, creating conditions conducive to developing diseases due to leaf wetness.

The highest leaf wetness level was recorded on July 13 for approximately 24 h. During this period, leaf wetness was measured at 568 h in the first year and 376 h in the second year. The total rainfall and leaf wetness period until July 20, when the disease symptoms first appeared, was 245 mm and 207 h, respectively, while in the second year, it was 162 mm and 213 h in the first year. Wind speeds fluctuated between 1.3 and 10.9 km h$^{-1}$, generally correlating with the number of rainy days ($R^2 = 0.40$).

CLS conidia germinate on the leaf surface and penetrate the pores, known as stomata. Symptoms of infection typically develop 5 to 21 days later, depending on weather conditions. The daily infection value (DIV) model tracks the number of hours with relative humidity above 85% or the presence of leaf wetness. Depending on the sensitivity of the crop variety, a DIV of six or more or a risk value of three or more for two consecutive days indicates the need for the first fungicide application (Fig. 6). In cases where disease-resistant varieties are available, higher DIV values can be expected over consecutive days (*Metos, 2024*).

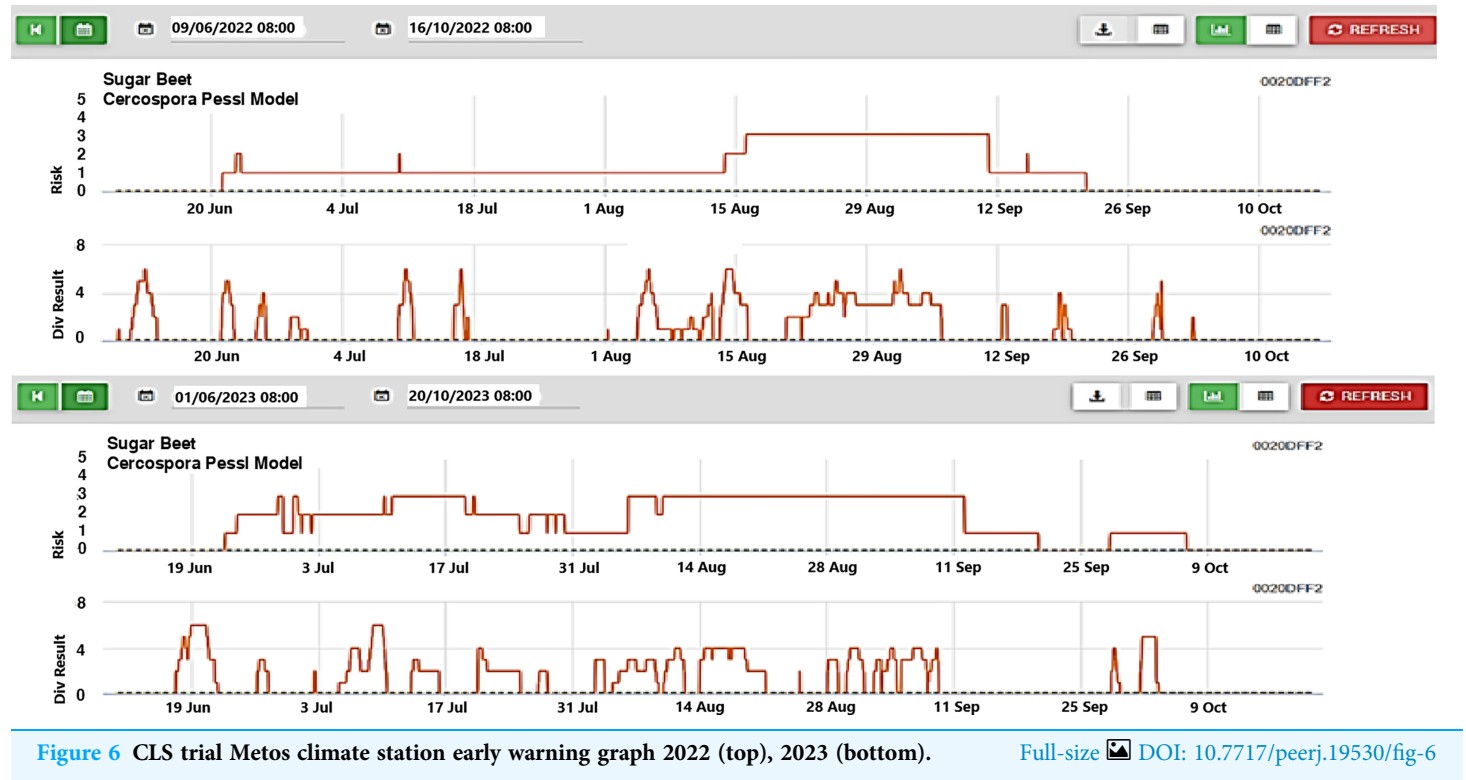

**Figure 6** CLS trial Metos climate station early warning graph 2022 (top), 2023 (bottom).

Between July and October, when the research was conducted in the PM field, the daily average air temperature ranged from 9 to 29 °C, while relative humidity varied between 35% and 89%. The air temperature remained relatively stable until September in the first year and August in the second year. Following this period, an inverse relationship was observed, where relative humidity increased in a fluctuating pattern. Although precipitation levels were either zero or extremely low on most days during the specified period, 0 to 9 mm of rainfall was recorded on August 14–16. Wind speed varied from 1.3 to 5.8 km h$^{-1}$, showing no correlation with the amount of rainfall, and remained consistent throughout the study.

## Image acquisition and analysis
### CLS trial

The acquisition of images and ground measurements for the CLS trial began on June 16, 2022, and continued until June 17, 2023, in the Yenişehir-Karaköy area, depending on the sugar beet's development status. Images were captured at intervals of 5 to 10 days, based on the region's climatic conditions and the processing time required for the images. A total of 11 observations were made during the first year, while 10 observations were recorded in the second year at the CLS site.

The index graphics is an effective tool for visualizing the health of sugar beet fields, designed explicitly for the plots on the DSM maps. In the index maps generated from MS images, green indicates areas where the sugar beet plants are healthy. In contrast, red highlights regions where disease is worsening, marked by an increase in spots on the leaves.

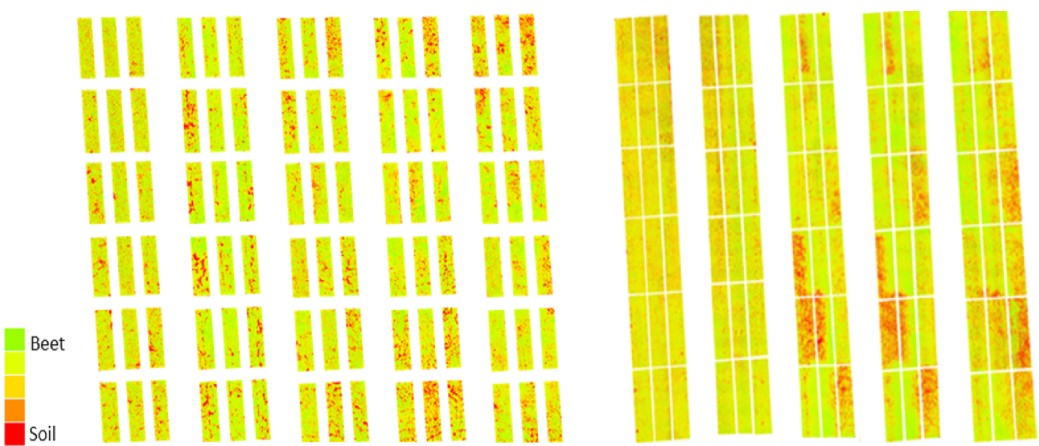

**Figure 7** MS images were taken in the CLS area on 15, 30 July, 18, 25 August, and 14 September in 2022, and 23 June, 15, 22 July, and 5, 17 August in 2023, respectively.

As the disease progresses, the healthy green color of the leaves changes to orange, revealing gaps and soil where the initial leaves have died (Figs. 2, 7).

The graph illustrating the index values obtained during the research shows distinct green bands representing healthy plant areas and red bands indicating soil. These bands are crucial for monitoring physiological development and disease presence (Fig. 8). In the MS images, increases in the green band reflect enhanced vegetation while decreasing signal plant stress. In the first year of the study, significant changes began on June 10, when a mild to moderate infection warning (div:6) was issued, lasting until June 13, June 22, and July 2. Risk values escalated to mild between June 21 and August 13, then to medium (div:2) on August 14, and reached severe (div:3) on August 16. Severe warnings persisted from August 16 to September 11, followed by a decrease in warnings from September 12 to September 21, after which no further risk warnings were issued. According to the div values, mild warnings fluctuated between August 15 and September 7, with additional mild warnings issued on September 13, 19, 20, and 30. As temperatures dropped in October, the significance of these warnings diminished (Fig. 8).

Several factors can contribute to plant stress. Observations of the parcels on July 3, 2022, indicated that the decline in indices was primarily caused by PM. Although leaf wetness was expected to occur around this time based on long-term data, the high day-night temperatures, paired with average relative humidity levels remaining around 70%, created an environment conducive to PM. After this period, leaf wetness values fluctuated due to intermittent rainfall, but it is believed that conditions suitable for the development of CLS had not yet been established. As of July 11, 2022, the irrigation div value reached six, and the risk indicator rose to two. Consequently, the S2 treatment was applied on July 12, 2022. By July 15, 2022, the impact of PM had decreased, and the index improved, likely due to temperature and relative humidity changes. Despite around 130 mm of rainfall occurring between July 14 and 17, the risk indicator remained unchanged, though the div value increased to six on July 18, 2022. Furthermore, the MS green band index value declined in

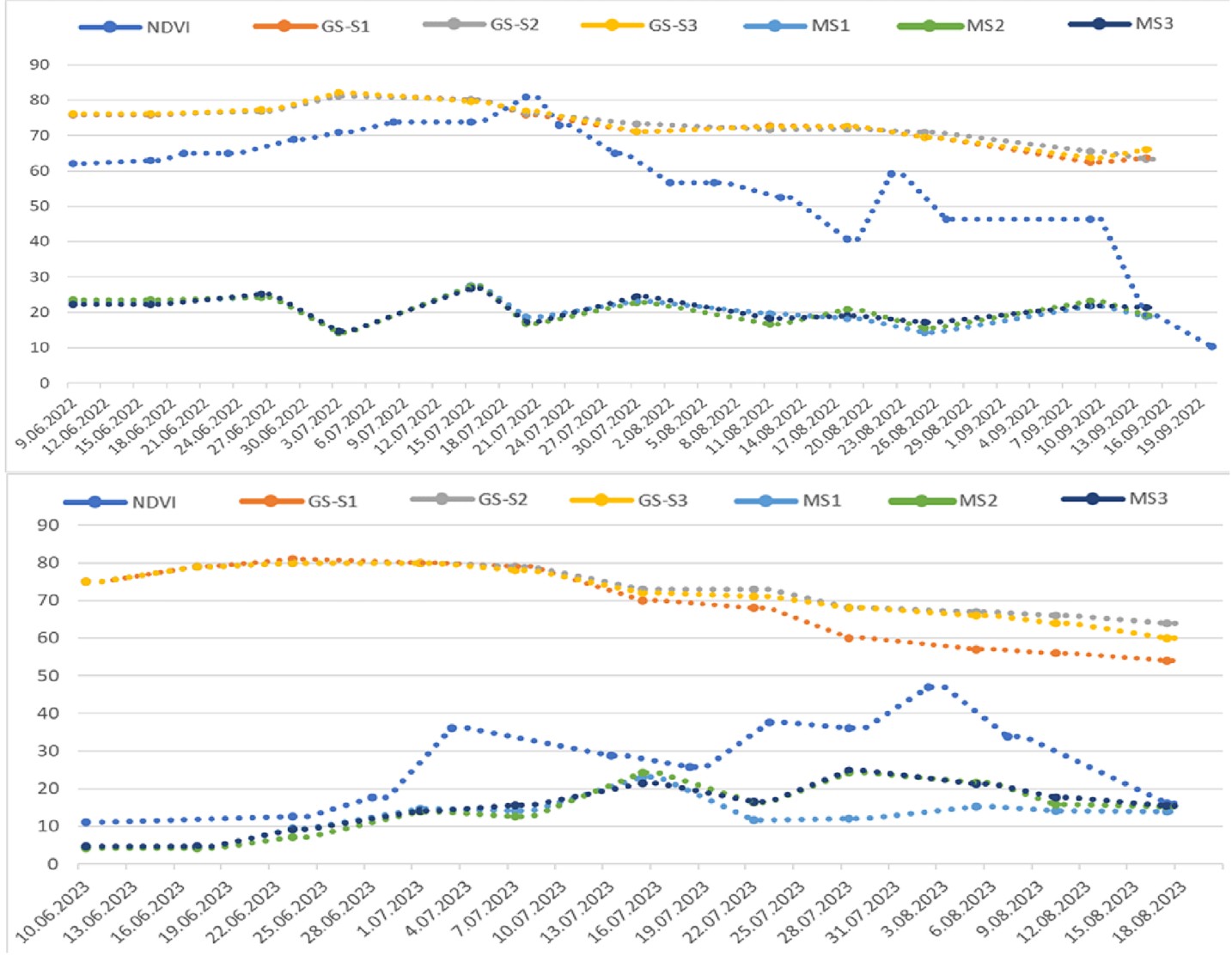

**Figure 8 Sentinel 2A satellite NDVI values taken from the CLS field in 2022 (top) and 2023 (bottom) and MS index values taken with GreenSeeker (GS) and drone (MS) according to the trial plots.**

images taken on July 20, 2022. Humidity levels remained stable throughout this period while a slight upward temperature trend existed.

The first spraying of the S3 subjects was conducted on July 24, 2022, in response to the appearance of one or two spots on the lower leaves that were first noted during field assessments. This observation and a decline in the index value indicated that the disease might be starting. In the following period, spraying continued at 15 to 20-day intervals on the S2, where pesticides were applied based on Metos data. While care was taken to maintain the spraying intervals for S3, fluctuations in index values were observed after the applications, possibly because of the fungicides (Fig. 8). Throughout the research, five sprays were conducted on the S2 subjects and four on the S3. Although the spraying was performed on scheduled dates, especially in early August, the first leaves began to die,

followed by the second and third. As illustrated in the images, despite a drop in index values on August 25, 2022, the index values later increased due to the loss of leaves. This increase can be attributed to the declining leaf density's intensified weed growth in the vacant areas, which raised the green band index in the drone images taken during this period.

In the second year of the experiment, the climate station issued its first disease warning on June 19. The S2 was sprayed when the risk indicator reached around two on June 22, 2023 (Fig. 6). Although the danger value (div) fluctuated intermittently after this date, the risk value remained relatively stable at levels 2–3 until August 20, which allowed the disease to develop. Approximately 70 mm of rainfall was recorded between July 7 and 10, causing the div value in the climate station's risk indicator to rise to 6. Consequently, the S2 was sprayed for the second time on July 9. Based on drone images under field conditions, the first decrease in index values was observed on July 3. Following field assessments that revealed spots on the first developing lower leaves of a few beets, spraying was conducted on the S3 on July 9.

Irrigation from July 19 to 21 increased leaf wetness. Despite the low div value, the risk value remained at two, and there was a decline in the MS index value. Therefore, the third spraying was carried out on S2 on July 21 and the second on S3 on July 24. Risk assessments indicated mild warnings from July 12 to 19, but the risk remained severe (div:3) after August 7. The risk value increased to level 3 due to rainfall between July 31 and August 1, followed by further irrigation. As a result, the fourth spraying was conducted on S2 on August 7 and the third on S3 on August 10. Additional spraying occurred on August 22 and 26. The spraying on S2 began on June 17 and continued six times at intervals of 15 to 20 days until September 15 (Fig. 9). For S3, where the disease was detected by drone, spraying was initiated 15 days later than planned in 2023, and from that point, the parcels were sprayed five times until September 15.

Overall, an evaluation of the drone images indicates that index values can effectively diagnose and detect the disease before it manifests.

### PM trial

The trial for monitoring PM imaging and measurements began in the areas on July 30, 2022, and August 1, 2023. During the study, air temperatures ranged between 25 °C and 30 °C, while relative humidity varied from 21% to 72%. Due to the lack of rainfall during this period, neither rainfall nor leaf wetness values were recorded. PM disease tends to manifest after the maturation of sugar beets, unlike CLS, which appears earlier in development. Additionally, the intensity of PM disease fluctuates based on climate and irrigation conditions, making monitoring more challenging (Fig. 9). Despite an increase in PM disease in the first year of the study, the data showed that index values remained relatively stable and showed minimal variability through 2022. Eight observations were made in the study's first year and seven in the second year.

The MS green band exhibited the most similar values regarding the disease progression. The index, which was notably high on July 30, 2022, decreased by August 11, 2022. The first spraying occurred simultaneously for subjects S2 and S3 on August 12, 2022.

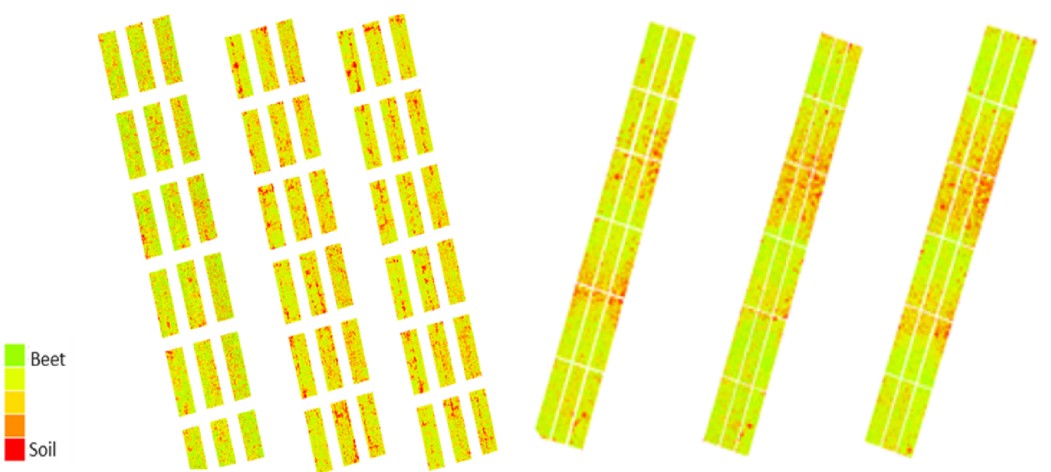

**Figure 9 MS images were taken in the PM area on August 11, 23, and September 3 in 2022, and August 1, 14, and 21 in 2023, respectively.**

Following this spraying, the index increased but dropped again 24 days later, on September 3, 2022, leading to a second spraying on September 5, 2022. After another decline on September 23, 2022, the final spraying was conducted on September 24, 2022. However, high weed growth in the Karacahöyük field during the first year of the research complicated the establishment of a clear relationship among all bands concerning disease development levels (Fig. 10).

The evaluations for PM in 2023 began on August 1 in the Eskişehir-Karagözler field. An increase in the index values from MS images taken from the field in the green band was observed until August 14. However, a decrease in the index was noted on August 21. In the second year of the research conducted in the PM field, only 0.2 mm of rainfall was recorded on August 14. Nevertheless, irrigation was performed weekly using a linear motion sprinkler system between July 3 and September 5. During this period, the daily average temperature ranged from 20 °C to 30 °C, and leaf wetness remained high due to the regular irrigation. This consistent wetting of the leaf surface also contributed to the washing away PM factors.

When the MS images were evaluated, the index drop on August 21 was interpreted as the onset of PM on the plants (Fig. 10). The first symptoms of PM were observed on the leaves on August 24. Based on the data collected, treatments S2 and S3 were applied on August 27. Since the progression of PM is influenced by climatic conditions and the trial land was irrigated systematically with a mobile irrigation system, it was possible to closely monitor the disease's development under controlled conditions. In the MS images taken on August 28, the horizontal trend continued after the spraying, although a further decrease was recorded in each band on September 4. Observations made following this decrease indicated that the disease level began to rise again. The rate of disease development after the initial and subsequent drops in index values supported the conclusion regarding the correlation between these drops and the onset of the disease. Consequently, two more treatments were carried out on September 10 and September 26.

![PeerJ]

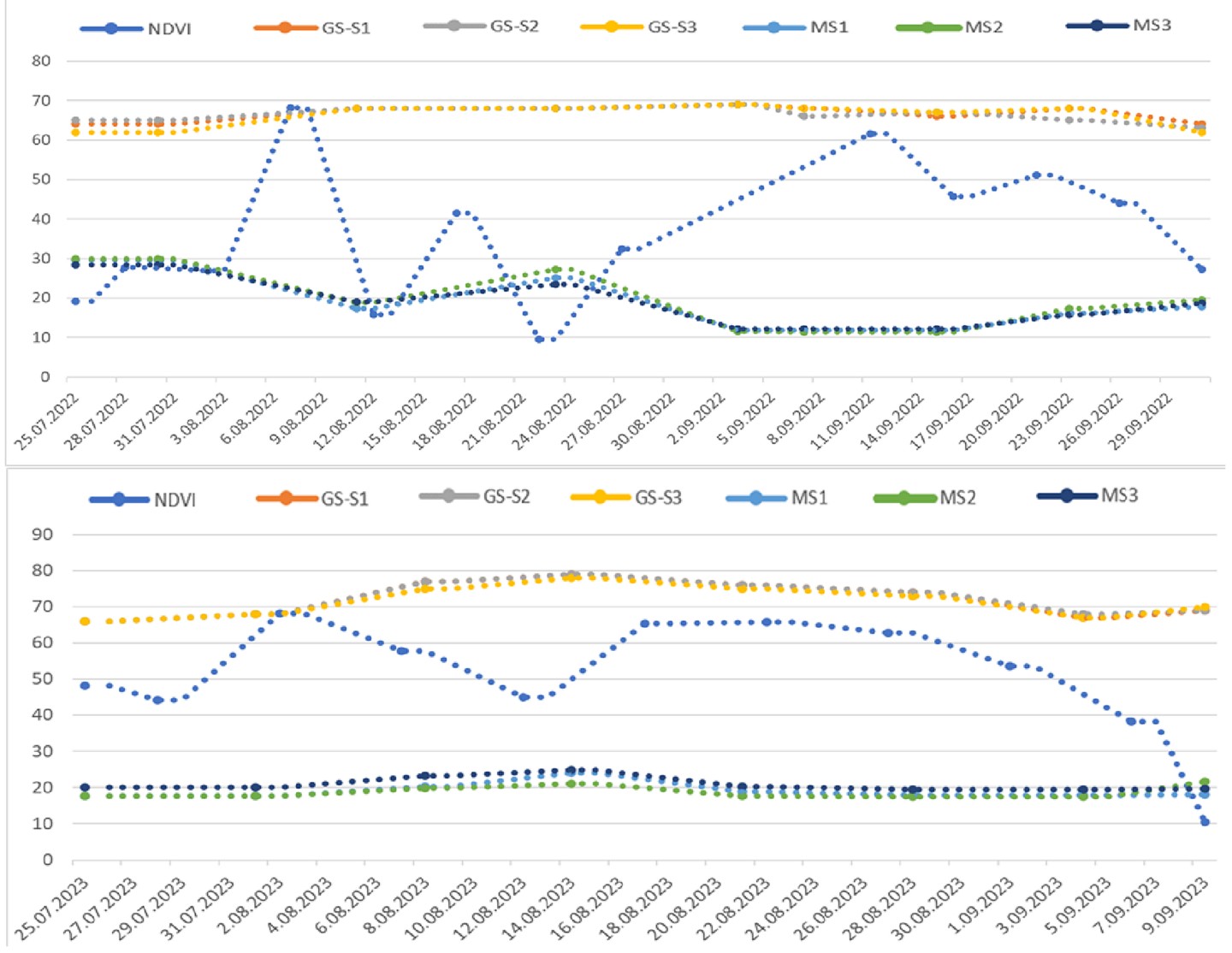

**Figure 10 Sentinel 2A satellite NDVI values taken from the PM field in 2022 (top) and 2023 (bottom) and MS index values taken with GreenSeeker (GS) and drone according to the trial plots.**

## Satellite images

Drone imaging provides more cost-effective options than satellite imaging systems, especially for small-scale projects. While drones with GPS and other sensors can obtain high-resolution images at low altitudes, they face challenges such as limited flight time and range and susceptibility to adverse weather conditions, making capturing observations over large areas difficult. For this reason, we evaluated 10-m resolution Sentinel 2A satellite images and NDVI maps provided by METOS for the same region to see if we could make regional predictions without visible disease symptoms using satellite imagery.

In the CLS experimental area, vegetation emerged around May 5 in the first year. It increased until July 8, peaked on July 21, and declined. Notably, the high values recorded

between July 5 and 20 and the decreases noted between August 2 and 14 are particularly striking in the graph. The graph of CLS field Sentinel 2A satellite images only partially aligns with the drone data due to differences in image acquisition dates. Still, both provide important information for monitoring vegetation changes. The declines observed on August 2, 17, 26, and September 10 indicate stress in the plants on those dates. It is particularly noteworthy that the differences of just a few days between the decreases observed in the index values from satellite and drone images (Fig. 8).

The analysis of NDVI values from the Sentinel 2A satellite for the PM field in 2022 reveals a dynamic pattern in vegetation changes. This data is essential for assessing the health and stress levels of the vegetation in the field. The development of sugar beet leaves in the PM field provides insight into the vegetation's status. The NDVI chart illustrates this pattern, showing a decline in maximum values around August 8–22 and September 11. Notably, the initiation of spraying on August 12, as indicated by drone images, corresponds with this trend, further supporting the analysis (Fig. 10).

In the second year of the research, it was noted that sugar beet leaf development in the CLS field began in May, consistent with the first year, and continued until the end of September, after which it started to decline. The Sentinel 2A satellite images showed a graph with index values in the 0.7–0.8 range, indicating leaf development. The first decrease in the index was observed between July 15 and 18, followed by a second decrease on July 28. A continuous decline, indicating a permanent reduction in vegetation, was noted from August 2 until August 18. In contrast, the index graphs derived from drone images showed the first decrease on July 7 and the second on July 22 (Fig. 8). When comparing the two methods, there was a 1-week discrepancy between the satellite and drone image indexes. Although this timing difference could pose challenges for early disease diagnosis and control, it remains significant for monitoring crops over large areas.

The chart also showed a decline in average and maximum NDVI values at the end of July and mid-August, specifically on July 28 and August 12, as indicated in the index chart. The first decrease in the drone image index was noted on August 21 (Fig. 10). In the second year, following a significant difference between the two methods applied in the PM field, the first occurrence of the disease observed in the field was identified after August 24, corresponding with the drone imagery findings.

## Terrestrial measurement

According to the data obtained from GS for 2022, NDVI values began to increase on June 16, reaching their peak on July 3. After this date, the values decreased for all three subjects until July 30. Between July 30 and August 18, the NDVI values stabilized, but a second decrease occurred after August 18. The GS measurements do not align with the drone images because the device consistently evaluates a specific detection area with each measurement. According to Metos climate data, the first spraying occurred on July 12, while drone images indicate that spraying occurred on July 24. This discrepancy suggests that the decline in NDVI values observed after July 3 in the GS measurements may be attributed to regional variations (Fig. 8).

In the NDVI chart obtained from GS measurements in the PM field, there is a notable correlation between the downward trend in the data and the pesticides applied after August 11, as well as on September 3 and September 23. This relationship is significant for understanding plant health. Leaf development in sugar beets peaks around mid-July, and although the number of leaves begins to decline and levels off until mid-August, root development continues (Fig. 10). Drone-based multispectral images taken in the region on August 11, 2022, decreased the NDVI index value. The first spraying occurred on August 12, 2022, after which disease development was observed. Given the lack of rainfall and the stage of leaf development during this period, any increases in the NDVI values may result from a reduction in disease severity attributed to irrigations performed on August 25 and September 23. Therefore, disease progression can be effectively monitored using GS data.

According to data collected in 2023 using GS, NDVI values have increased since the initial measurement on June 10. After peaking on July 1, these values began to decline until the final measurement date on August 17 for all three subjects. While GS measurements do not precisely correlate with drone images, a notable alignment on July 7, the date of the first observed fracture, coincides with when the drone detected the disease (Fig. 8). Following August 14, NDVI values measured by GS decreased, and a relationship was noted between the GS data and the appearance of disease symptoms.

At the onset of the disease in both regions, the GS values closely align with the drone images. The minor discrepancies observed in subsequent periods can be attributed to the differing measurement methods: drone images are area-based, while GS measurements are plant-based. Despite this difference, our data highlight the potential of local measurements for disease detection (Fig. 10). This finding emphasizes the practical significance of our research in precision agriculture.

## Field trials

### CLS trial

The disease was detected using Metos software and drone data 15 days earlier than usual, allowing early spraying to commence. Throughout the 2022 season in the CLS field, S2 received one more spray than S3. After the last spraying, the damage levels of the disease were evaluated using a 0-9 indicator, resulting in scores of 8.83, 6.83, and 7.33 for S1, S2, and S3, respectively. Significant differences were observed among these scores. According to the data from 2023, the damage levels after the last spraying were 9.00 for S1, 5.93 for S2, and 6.47 for S3, again showing significant differences among them (Table 2).

The disease rates recorded were 98.15% in S1, 64.81% in S2, and 70.37% in S3 for 2022. The effectiveness of spraying was measured at 35.19% in S2 and 29.63% in S3. In the second year of the research, the disease rates were 98.15% in S1, 65.83% in S2, and 71.85% in S3. In 2023, like 2022, the disease rate was significantly higher with the drone method than with the Metos software, which was attributed to the late detection of CE spots. The effectiveness of spraying for that year was found to be 34.17% in S2 and 28.15% in S3 (Table 3).

Research conducted during the CLS trial in 2022 and 2023 revealed statistically significant beet yield and sugar yield differences among S1, S2, and S3 ($P < 0.01$). However,

**Table 2 CLS disease observation results according to the 0-9 scale.**

| Method | 31.08.2022 | | | | | | | 14.09.2022 | | | | | | | 23.09.2022 | | | | | | |
|---|---|---|---|---|---|---|---|---|---|---|---|---|---|---|---|---|---|---|---|---|---|
| | I | II | III | IV | V | VI | Mean | I | II | III | IV | V | VI | Mean | I | II | III | IV | V | VI | Mean |
| S1 | 6 | 6 | 6 | 5 | 7 | 6 | 6.00 | 8 | 8 | 8 | 8 | 7 | 8 | 7.83 | 9 | 9 | 9 | 9 | 8 | 9 | 8.83 |
| S2 | 3 | 3 | 3 | 3 | 3 | 4 | 3.17 | 5 | 5 | 5 | 5 | 4 | 5 | 4.83 | 7 | 7 | 7 | 7 | 6 | 7 | 6.83 |
| S3 | 5 | 4 | 5 | 5 | 5 | 5 | 4.83 | 6 | 6 | 7 | 7 | 6 | 6 | 6.33 | 7 | 7 | 8 | 8 | 7 | 7 | 7.33 |
| | 15.07.2023 | | | | | | | 28.07.2023 | | | | | | | 10.08.2023 | | | | | | |
| S1 | 3 | 6 | 3 | 5 | 7 | 5 | 4.83 | 7 | 8 | 7 | 7 | 7 | 8 | 7.33 | 9 | 9 | 9 | 9 | 9 | 9 | 9.00 |
| S2 | 2 | 2 | 2 | 2 | 2 | 2 | 2.17 | 4 | 5 | 4 | 5 | 4 | 4 | 4.33 | 6 | 6 | 6 | 6 | 5 | 7 | 5.93 |
| S3 | 2 | 2 | 3 | 3 | 5 | 4 | 3.17 | 4 | 4 | 5 | 5 | 5 | 5 | 4.67 | 6 | 7 | 7 | 6 | 7 | 7 | 6.47 |

**Table 3 Townsend-Heuberger disease rates and Abbott pesticide activities in CLS plot.**

| Method | Disease rate (%) | | | | | | | Effectiveness of pesticides (%) | | | | | | |
|---|---|---|---|---|---|---|---|---|---|---|---|---|---|---|
| 2022* | I | II | III | IV | V | VI | Mean | I | II | III | IV | V | VI | Mean |
| S1 | 100.0 | 100.0 | 100.0 | 100.0 | 88.89 | 100.0 | 98.15a | | | | | | | |
| S2 | 66.67 | 66.67 | 66.67 | 66.67 | 55.56 | 66.67 | 64.81c | 33.33 | 33.33 | 33.33 | 33.33 | 44.44 | 33.33 | 35.19 |
| S3 | 66.67 | 66.67 | 77.78 | 77.78 | 66.67 | 66.67 | 70.37b | 33.33 | 33.33 | 22.22 | 22.22 | 33.33 | 33.33 | 29.63 |
| 2023** | I | II | III | IV | V | VI | Mean | I | II | III | IV | V | VI | Mean |
| S1 | 100.0 | 100.0 | 100.0 | 100.0 | 88.89 | 100.0 | 98.15a | | | | | | | |
| S2 | 65.56 | 67.78 | 64.44 | 63.33 | 56.67 | 77.22 | 65.83c | 34.44 | 32.22 | 35.56 | 36.67 | 43.33 | 22.78 | 34.17 |
| S3 | 66.67 | 72.22 | 75.56 | 68.89 | 73.33 | 74.44 | 71.85b | 33.33 | 27.78 | 24.44 | 31.11 | 26.67 | 25.56 | 28.15 |

Note:
* Observations in the CLS parcel were made on 23.09.2022 in the first year of the research.
** Observations in the CLS parcel were made on 10.08.2023 in the second year of the research.
Treatment means in table were compared using the Duncan multiple comparison method, and group means were ranked from largest to smallest as a, b, c.

**Table 4 CLS combined yield and quality analysis results.**

| Method | Beet yield t ha$^{-1}$ | Sugar content (%) | Sodium content Meq Na/100g | Potassium content Meq K/100g | Nitrogen content Meq N/100g | Refined sugar content (%) | Refined sugar content t ha$^{-1}$ |
|---|---|---|---|---|---|---|---|
| S1 | 107.48b | 10.24b | 3.68 | 4.42 | 3.82 | 6.81b | 7.29b |
| S2 | 120.88a | 10.74a | 4.40 | 4.50 | 3.68 | 7.34a | 8.83a |
| S3 | 118.33a | 10.48ab | 4.15 | 4.36 | 3.86 | 7.02ab | 8.23a |

Note:
Treatment means in table were compared using the Duncan multiple comparison method, and group means were ranked from largest to smallest as a, b.

the difference between subjects S2 and S3 was insignificant (Table 4). The sugar rates for the subjects were 10.24%, 10.74%, and 10.48%, respectively, with a significant difference noted ($P < 0.05$). Subject S2 had the highest sugar content, although the difference compared to S3 was not significant. The time lag between identifying the disease and the initiation of spraying in subjects S2 and S3 contributed to a partial decrease in the sugar rate. A similar pattern was observed in the refined sugar ratio.

Sugar yields varied by subject, measuring 7.29, 8.83, and 8.23 t ha$^{-1}$. There was no significant difference between subjects S2 and S3, but both differed significantly from

**Table 5 PM disease observation results according to the 0–5 scale.**

| Method | 31.08.2022 | | | | | | | 15.09.2022 | | | | | | | 23.09.2022 | | | | | | |
|---|---|---|---|---|---|---|---|---|---|---|---|---|---|---|---|---|---|---|---|---|---|
| 2022 | I | II | III | IV | V | VI | Mean | I | II | III | IV | V | VI | Mean | I | II | III | IV | V | VI | Mean |
| S1 | 3 | 3 | 3 | 3 | 3 | 3 | 3.00 | 4 | 4 | 3 | 4 | 4 | 4 | 3.83 | 4 | 4 | 4 | 5 | 4 | 4 | 4.17 |
| S2 | 3 | 2 | 3 | 2 | 2 | 2 | 2.33 | 3 | 2 | 2 | 2 | 3 | 2 | 2.33 | 3 | 2 | 3 | 2 | 3 | 3 | 2.67 |
| S3 | 3 | 2 | 3 | 2 | 2 | 3 | 2.50 | 3 | 2 | 2 | 2 | 2 | 2 | 2.17 | 4 | 3 | 3 | 2 | 3 | 2 | 2.83 |
| 2023 | 01.09.2023 | | | | | | | 15.09.2023 | | | | | | | 28.09.2023 | | | | | | |
| S1 | 2 | 3 | 3 | 1 | 1 | 1 | 1.82 | 2 | 3 | 3 | 4 | 4 | 2 | 3.01 | 5 | 4 | 5 | 5 | 5 | 5 | 4.84 |
| S2 | 1 | 1 | 1 | 0 | 0 | 0 | 0.52 | 0 | 0 | 1 | 1 | 1 | 0 | 0.50 | 2 | 2 | 3 | 2 | 3 | 3 | 2.50 |
| S3 | 1 | 1 | 1 | 0 | 0 | 0 | 0.50 | 0 | 1 | 0 | 0 | 1 | 0 | 0.33 | 2 | 3 | 2 | 2 | 3 | 2 | 2.34 |

subject S1 ($P < 0.01$). The combined results over the two-year study period showed no significant differences in sodium, potassium, or harmful nitrogen contents among the three trial subjects (Table 4).

*PM trial*

PM area was sprayed three times from the emergence of the crops to the harvest date in 2022. Physical observations were conducted using a 0–5 scale to assess the level of disease damage. The results indicated a damage level of 4.17 in S1, while S2 and S3 recorded damage levels of 2.67 and 2.83, respectively (Table 5). The difference between S2 and S3 was not statistically significant. In the second year of the study, three sprayings were also performed from crop emergence to harvest, like the first year. Evaluations conducted using the 0–5 scale revealed a damage level of 4.84 in S1, with S2 and S3 showing damage levels of 2.50 and 2.34, respectively (Table 5).

The observed disease rates in the trial were 83.77%, 53.40%, and 56.43% for the three groups, respectively. The effectiveness of the spraying treatments was 46.60% for S2 and 43.57% for S3. In the second year of the research, the disease rates were 96.73% for S1, 49.90% for S2, and 46.77% for S3. The effectiveness of the treatments for S2 and S3 was measured at 48.81% and 51.65%, respectively (Table 6).

According to the combined analysis results from two years of field trials for the PM trial, the root yields were 90.07, 101.06, and 100.05 t ha$^{-1}$, respectively. Although no statistically significant difference was found between S2 and S3, significant differences were observed between S2, S3 and S1 ($P < 0.05$). Similar results were noted for sugar yield, which measured 13.49, 15.26, and 15.39 tons, respectively. The combined results regarding PM disease damage indicated no differences among the methods regarding sugar content, refined sugar content, sodium, potassium, and harmful nitrogen levels (Table 7).

# MACHINE LEARNING ANALYSIS

Binary classification was performed on the CLS and PM disease datasets based on the presence or absence of disease information. The classification results were also assessed using heatmap graphics to visualize the similarities or differences in the dataset values.

This research utilized DSM data alongside the values obtained from the Pix4D Mapper program, specifically Green_MS, Yellow_MS, Orange1_MS, Orange2_MS, and Red_MS,

**Table 6 Disease rates and effectiveness of pesticide in PM plots.**

| Method | Disease rate (%) | | | | | | | Effectiveness of pesticides (%) | | | | | | |
|---|---|---|---|---|---|---|---|---|---|---|---|---|---|---|
| **2022** | I | II | III | IV | V | VI | Mean | I | II | III | IV | V | VI | Mean |
| S1 | 82.20 | 79.20 | 80.20 | 99.80 | 79.60 | 81.60 | 83.77a | | | | | | | |
| S2 | 58.20 | 40.20 | 59.60 | 42.40 | 60.60 | 59.40 | 53.40b | 41.80 | 59.80 | 40.40 | 57.60 | 39.40 | 40.60 | 46.60 |
| S3 | 79.80 | 60.40 | 59.80 | 39.80 | 58.20 | 40.60 | 56.43b | 20.20 | 39.60 | 40.20 | 60.20 | 41.80 | 59.40 | 43.57 |
| **2023** | I | II | III | IV | V | VI | Mean | I | II | III | IV | V | VI | Mean |
| S1 | 99.20 | 81.80 | 99.60 | 100.00 | 99.80 | 100.00 | 96.73a | | | | | | | |
| S2 | 42.00 | 41.60 | 57.20 | 40.40 | 58.80 | 59.40 | 49.90b | 56.58 | 56.99 | 40.87 | 58.23 | 39.21 | 38.59 | 48.41 |
| S3 | 43.00 | 55.80 | 41.80 | 43.00 | 55.60 | 41.40 | 46.77b | 55.55 | 42.31 | 56.79 | 55.55 | 42.52 | 57.20 | 51.65 |

Note:
Treatment means in table were compared using the Duncan multiple comparison method, and group means were ranked from largest to smallest as a, b.

**Table 7 PM combined yield and quality analysis results.**

| Method | Beet yield t ha$^{-1}$ | Sugar content (%) | Sodium content Meq Na/100 g | Potassium content Meq K/100 g | Nitrogen content Meq N/100 g | Refined sugar content (%) | Refined sugar yield t ha$^{-1}$ |
|---|---|---|---|---|---|---|---|
| S1 | 90.07b | 18.01 | 1.03 | 5.59 | 3.57 | 15.11 | 13.49b |
| S2 | 101.06a | 18.11 | 0.92 | 5.56 | 3.42 | 15.27 | 15.26a |
| S3 | 100.05a | 18.33 | 0.88 | 5.44 | 3.27 | 15.57 | 15.39a |

Note:
Treatment means in table were compared using the Duncan multiple comparison method, and group means were ranked from largest to smallest as a, b.

**Table 8 CLS disease machine learning analysis results.**

| Year | Method | Confusion matrix | Accuracy | Precision | Recall | F1-score |
|---|---|---|---|---|---|---|
| 2022–2023 combined | LR | [[0 159]] [0 81]] | 0.71 | 0.54 | 1.00 | 0.70 |
| | KNN | [[132 27] [15 66]] | 0.83 | 0.71 | 0.81 | 0.76 |
| | DTC | [[100 59] [0 81]] | 0.72 | 0.58 | 1.00 | 0.73 |
| | SVM | [[109 50] [3 78]] | 0.77 | 0.62 | 0.96 | 0.75 |
| | GNB | [[100 59] [0 81]] | 0.72 | 0.58 | 1.00 | 0.73 |

for CLS. The results indicated that the KNN algorithm achieved the highest accuracy and F1 score in analyzing multispectral (MS) images (Table 8).

The heat map is a tool used to evaluate research results by displaying the values of the main variable studied in a grid of colored squares across two axes. Each cell's color represents the value of the main variable within a specific range. Heat maps illustrate the relationship between two variables by plotting one variable along each axis. The color gradient ranges from light to dark, indicating the strength of the relationship between the variables. A correlation approaching 1 signifies a strong relationship, facilitating the
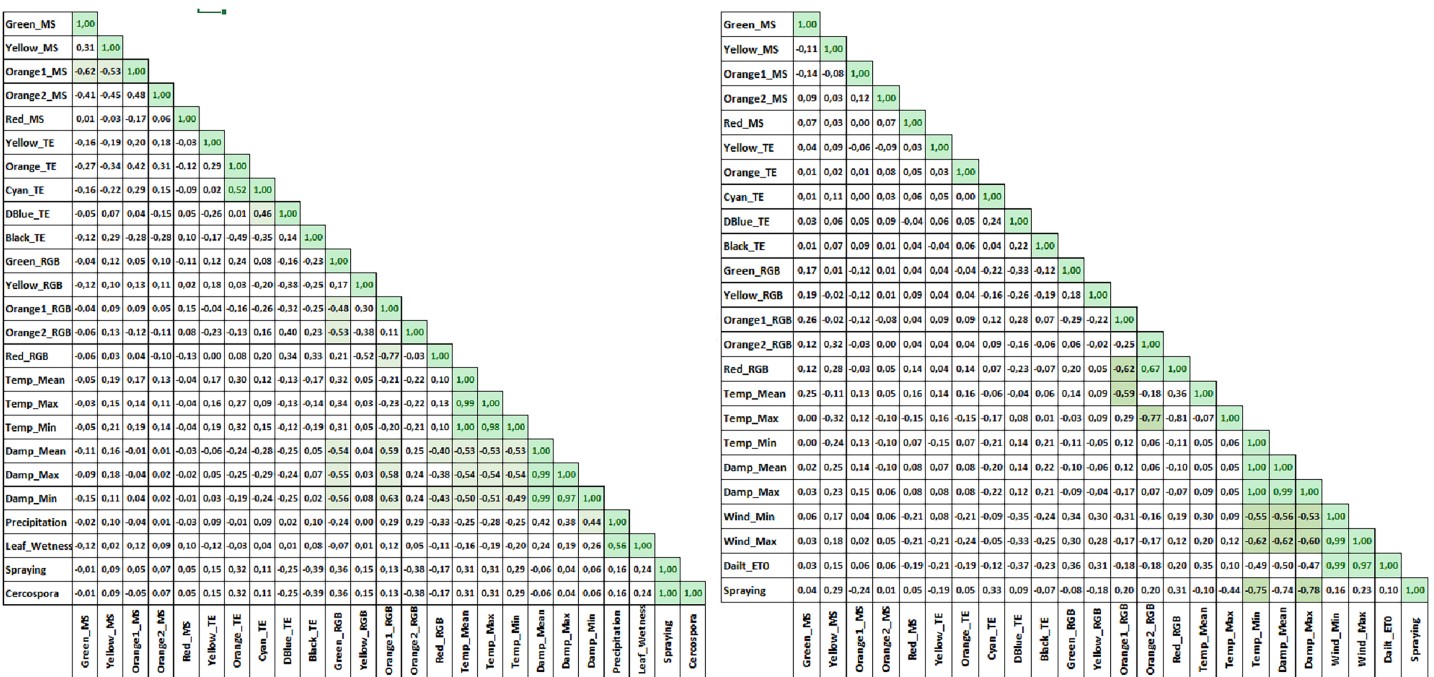

**Figure 11 Combined CLS heat map graph (left) and PM heat map graph (right).**

**Table 9 PM disease, machine learning analysis results.**

| Year | Method | Confusion matrix | Accuracy | Precision | Recall | F1-score |
|---|---|---|---|---|---|---|
| 2022–2023 combined | LR | [[40 1]<br>[10 69]] | 0.91 | 0.99 | 0.87 | 0.93 |
| | KNN | [[36 5]<br>[10 69]] | 0.88 | 0.93 | 0.87 | 0.90 |
| | DTC | [[36 5]<br>[10 69]] | 0.88 | 0.93 | 0.87 | 0.90 |
| | SVM | [[36 5]<br>[10 69]] | 0.88 | 0.93 | 0.87 | 0.90 |
| | GNB | [[31 10]<br>[12 67]] | 0.82 | 0.87 | 0.85 | 0.86 |

interpretation of the primary variable under investigation. This visualization provides a more detailed understanding of the sub-factor's influences on the studied primary variable. In the heat map graphics, temperature and humidity parameters are identified as the primary factors in the formation of CLS disease (Fig. 11).

The MS data analysis for PM disease was conducted using values from the DSM data, specifically the Green_MS, Yellow_MS, Orange1_MS, Orange2_MS, and Red_MS. The results for accuracy and F1-score from the MS images indicated that the LR algorithm outperformed the others (Table 9). The accompanying graph demonstrates that temperature and humidity are significant factors in diagnosing the disease (Fig. 11). These

findings suggest that early detection of CLS and PM diseases is achievable through analyzing the generated dataset using KNN and LR algorithms.

## DISCUSSION

Sugar beet is an important part of human nutrition, a significant agricultural product, a source of bioenergy, and an industrial raw material. Although sugar beet is widely cultivated, particularly in the sugar industry, it suffers from substantial yield losses due to various pests and diseases. Implementing effective control measures for these issues is vital for enhancing agricultural production sustainably and efficiently. This research aims to detect CLS and PM diseases at an early stage using drone-based remote sensing technology, highlighting both the feasibility and advantages of this method.

This research utilized LR, KNN, GNB, SVM, and DTC as machine-learning algorithms. The study focused on determining the presence or absence of the disease through binary classification of the dataset. Analyses were conducted on 16,440 data points for PM and 19,140 for CLS. Climate data, including average air temperature, relative humidity, rainfall amounts, and wind speed for the region during two periods, were considered when evaluating CLS disease. July and August were the months with the highest temperatures in the CLS field, while values for leaf wetness and relative humidity varied depending on rainfall and irrigation amounts. During July and August, the development period for sugar beet, the region experienced low overall rainfall, interrupted by periods of heavy rain. Leaf wetness, resulting from either rainfall or irrigation, combined with suitable temperatures, is a key factor in the emergence and development of CLS disease (*El Jarroudi et al., 2021*).

In contrast, a drier climate was observed in the PM area, even though the temperature values were like those in the CLS area. Like CLS disease, air temperature plays an important role in the onset and progression of PM disease. Color intensity changes on the DSM and index maps derived from MS images captured by drones at regular intervals during the plant's development indicate a relationship with the stress state of sugar beets. This research involved monitoring plant physiology and disease status by analyzing graphs of index values obtained from processed drone images. The green band, particularly evident in the MS images, has been identified as a key indicator of plant development and stress conditions (*Pix4D, 2023*). While there is extensive research on the remote sensing detection of CLS disease after its symptoms appear, as well as studies on its intensity and developmental status (*Rumpf et al., 2010*; *Görlich et al., 2021*), there is limited research on the ability to determine whether the disease will occur before physical symptoms are present.

PM disease in sugar beet typically occurs after the crop has matured, and its severity can change rapidly based on climate and irrigation conditions. This variability complicates early detection (*Awad et al., 2015*). PM disease manifested during both years of the study, in July and August. It was attributed to high temperatures, low rainfall, and humidity conditions that correspond with the ripening of the beets. Despite this, the disease index values remained relatively stable over time. This stability is likely attributed to planned irrigation practices, which helped wash away PM spores and manage the disease's

progression. The breaks observed in the MS green band index map also provided valuable insights for tracking the disease's development. The findings from the PM field studies suggest a correlation between the breaks in index values and the early onset of the disease before its visible appearance (*Ma et al., 2018*). The two-year research results show that the index values created by parallelism and drone images are decisive in early disease detection. When the data obtained from the images are evaluated together with climate data, it is seen that disease prediction can be made with high accuracy with the drone. According to Metos climate station disease warning data, higher yield and quality values were obtained for S2 sprayed in predicting disease occurrence.

Although drones have some disadvantages, such as limited flight times and susceptibility to weather conditions, they offer significant economic advantages compared to satellite imagery (*Kogut, 2022*). In the research areas, NDVI maps derived from the 10 m resolution Sentinel 2A satellite were compared with drone images. The satellite image data from both years can serve as indicators for assessing plant stress. However, satellite and drone data only partially overlap regarding the development of sugar beet in the CLS and PM fields (*Segarra et al., 2020*). The initial stress indicators in the graphs were noted on July 5 in the CLS field and August 11 in the PM field in 2022, which were crucial for plant stress analysis, as the drone images closely resembled these index graphs. Changes in NDVI index values ranging from 0.7 to 0.8 in the satellite data provide meaningful insights into plant stress situations. While satellite images allow for the analysis of larger areas in a single capture, their 10m resolution limits evaluations compared to the higher resolution of drone images. Although high spatial and temporal resolution images are obtainable, they are often expensive, and deciding on the optimal dates for capturing images to provide early warnings presents a significant challenge (*ListenField, 2023*). Conversely, the 0.4 cm resolution used for the drone images in this research provides more sensitive data for tracking plant stress, which cannot be observed at 10 m resolution. Additionally, variations in temporal resolution and decreased image quality in cloudy weather can also affect the sensitivity of satellite-based evaluations (*Kogut, 2022*).

The ground-based measurements (GS) taken in the CLS field were compared with the indices from drone images. Although the overall trend of the terrestrial NDVI values does not perfectly align, a notable correlation exists on July 3, when the first significant break occurred (*Anderson et al., 2016*). The findings from the second year of research mirrored those of the first year concerning this initial fracture. Beyond this first break, the two methods have no significant similarity. This discrepancy arises because GS measurements are taken within the device's detection area, focusing on the same row within the parcels, and thus only represent a limited detection zone.

In contrast, drone images encompass the entire parcel (*Jin & Eklundh, 2014*). Consequently, regional variations within the field have contributed to the GS measurements not aligning fully with the drone images. However, similar relationships between terrestrial and aerial images were observed in the PM field.

Significant differences were observed in disease detection and spraying strategies due to the field applications of the research. In the CLS field, there is approximately a 15-day difference in the timing of spraying beginning between the S2 and S3 subjects in both

years. Starting the spraying later on S3 resulted in one less application than S2, which had positive outcomes. This delay not only reduced the total labor and chemical costs associated with pesticide application by about $43 per hectare, but it also contributed positively to environmental health by minimizing the use of chemicals. However, when looking at CLS disease observations, the disease rate for S2 was 8% lower, and the spraying effectiveness was 17% higher than for S3. Despite these differences in disease rates and spraying effectiveness, the impact of these applications on yield and quality parameters in the CLS field was limited. The beet yield, sugar content, and sugar yield were lowest in the S1 subject, while significant differences were noted between the S2 and S3 subjects ($P < 0.05$).

This research emphasizes that spraying before the visible onset of the disease can conserve time, labor, and chemicals without sacrificing productivity and quality in the battle against CLS. Regarding PM, there was no notable difference in the timing of when spraying commenced between the S2 and S3 subjects over both years. Given that the disease evolves rapidly due to climatic changes and that pesticides were applied immediately after the first symptoms were detected, the disease rate in S2 was 6% lower than in S3. The effectiveness of spraying was also 6% higher. The results indicate that the spraying strategies against PM disease do not significantly differ in root and sugar yields. Additionally, drone technology can be utilized as an effective tool in controlling the disease.

Data from Green_MS, Yellow_MS, Orange1_MS, Orange2_MS obtained from MS images were analyzed alongside Temperature_Avg, Humidity_Avg, Precipitation, and Leaf_Wetness data. After evaluating this data in the CLS field, both the KNN and LR algorithms demonstrated high accuracy during the initial stage, particularly when assessing the color data from the MS images regarding accuracy and F1-score metrics. According to previous research (*Hallau et al., 2018*; *Kamilaris & Prenafeta-Boldú, 2018*), KNN and SVM algorithms yield successful results in machine learning studies. Notably, it has been found that air temperature, humidity, and leaf wetness are crucial factors in enhancing the accuracy of disease prediction.

## CONCLUSION

The findings of this research, which focuses on the early detection of diseases in sugar beet agriculture through drone and satellite imagery, indicate that this method can support sustainable production with high efficiency and quality. The research provides valuable insights for the early detection of CLS and PM diseases, leveraging the data sets generated by machine learning algorithms. KNN and LR have demonstrated the highest prediction accuracy among these algorithms due to their excellent discrimination capabilities.

Remote sensing technologies are powerful tools in sugar beet agriculture, significantly enhancing agricultural productivity, optimizing resource management, and promoting sustainable farming practices. The study results are strengthened by utilizing the power of drone imaging and analysis to detect diseases in sugar beet crops, ensure healthier yields, and optimize agricultural practices. Furthermore, integrating climate data with image

analysis can improve the success rate of early disease diagnosis. This integration allows for timely pesticide applications, leading to better disease control.

UAV-based remote sensing technologies have significant potential in the early diagnosis and management of diseases in sugar beet production. Consequently, processing drone-captured images with artificial intelligence technologies, such as machine learning, can help identify disease-related stress factors early, mitigate economic losses in agricultural production, and enhance overall efficiency. This approach also supports advancements in farm technologies aimed at sustainable production practices.

### Funding
This study was supported by Scientific and Technological Research Council of Turkey (TUBITAK) under the Grant Number 221O287. The funders had no role in study design, data collection and analysis, decision to publish, or preparation of the manuscript.

### Grant Disclosures
The following grant information was disclosed by the authors:
Scientific and Technological Research Council of Turkey (TUBITAK): 221O287.

### Competing Interests
The authors declare that they have no competing interests.

### Author Contributions
- Koç Mehmet Tuğrul conceived and designed the experiments, performed the experiments, analyzed the data, prepared figures and/or tables, authored or reviewed drafts of the article, and approved the final draft.
- Rıza Kaya conceived and designed the experiments, performed the experiments, analyzed the data, prepared figures and/or tables, authored or reviewed drafts of the article, and approved the final draft.
- Kemal Özkan conceived and designed the experiments, performed the experiments, analyzed the data, prepared figures and/or tables, and approved the final draft.
- Merve Ceyhan conceived and designed the experiments, analyzed the data, prepared figures and/or tables, and approved the final draft.
- Uğur Gürel conceived and designed the experiments, analyzed the data, prepared figures and/or tables, and approved the final draft.
- Fatih Yavuz Fidantemiz conceived and designed the experiments, performed the experiments, prepared figures and/or tables, and approved the final draft.

### Data Availability
The data and code are available at GitHub and Zenodo:
- https://github.com/merveceyhan/sugar_beet_classification_with_dataset.

- Ceyhan, M. (2025). Sugar Beet Dataset [Data set]. Zenodo. https://zenodo.org/records/15366412.

## Supplemental Information

Supplemental information for this article can be found online at http://dx.doi.org/10.7717/peerj.19530#supplemental-information.

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
