# Peer review of "Early detection of Cercospora beticola and powdery mildew diseases in sugar beet using uncrewed aerial vehicle-based remote sensing and machine learning"

_PeerJ, doi:10.7717/peerj.19530_

## Round 0.1 · original submission · Major Revisions

Cercospora leaf spot disease and powdery mildew are two significant fungal threats affecting sugar beet fields. Early detection of these pathogens is critical for implementing effective plant protection measures and minimizing losses in yield and quality. In this context, UAVs (Unmanned Aerial Vehicles) have emerged as promising tools for the rapid and accurate detection of these pathogens, offering a significant advantage over traditional diagnostic methods.

To enhance the clarity and impact of your article, it is essential to address specific technical aspects thoroughly. I strongly encourage a careful review of the reviewers' suggestions, thoughtfully evaluating each recommendation. If you disagree with any suggestion, it would be helpful to provide clear, well-reasoned justifications for your viewpoint.

Additionally, your article requires linguistic refinement. You might consider seeking assistance from a colleague with expertise in scientific editing or utilizing our professional editing service to achieve the highest standards of clarity and readability.

·

Basic reporting

Language and Clarity: The manuscript is generally well-written, but there are areas where clarity and professional tone can be improved. For instance:

Some sentences, such as those in the introduction and results sections, are overly complex. Simplifying the language could help enhance readability for a broader audience.
There are minor grammatical errors and awkward phrasing (e.g., lines 23–28). I recommend the authors have the manuscript reviewed by a professional language editor or a colleague proficient in English.
Literature References and Context:

The introduction provides a broad overview of the importance of remote sensing in agricultural management and disease detection. However, more recent and relevant literature could be cited to strengthen the background.
While the references used are appropriate, some critical recent studies on UAV-based disease monitoring techniques (e.g., from 2023) are missing and should be incorporated to contextualize the study better.
Article Structure, Figures, and Tables:

The article conforms to standard sections. The methods and results are detailed and logically organized. However, the formatting of the figures could be improved. For instance:
Figures 3 and 4 lack clear annotations, making it difficult for readers to understand the presented findings at first glance.
Figure captions should be more descriptive to provide a clear standalone explanation of what is being displayed.
The tables provide valuable data but would benefit from more consistent formatting and the inclusion of explanations for abbreviations and symbols in the footnotes.
Raw Data Sharing:

The authors have provided raw data files, which align with PeerJ's policy. However, the metadata or additional descriptions for the raw datasets could be more detailed, especially for readers unfamiliar with the specific tools or software used.
Relevance and Results:

The manuscript is self-contained, presenting all the necessary results to address the stated hypotheses. The results are comprehensive and well-integrated into the discussion.
Suggested Improvements:

Enhance clarity and language by revising complex sentences and addressing minor grammatical issues.
Expand the introduction to include recent advancements in UAV and machine learning for agricultural applications.
Improve the quality and annotations of figures and tables for better interpretability.

Experimental design

Original Primary Research and Research Question:

The research aligns well with the journal’s aims and scope, focusing on the integration of UAV-based remote sensing and machine learning to address a relevant issue in agricultural disease detection. The research question is clearly stated, and the study aims to fill an identified knowledge gap by demonstrating the application of spectral indices and machine learning for early detection of Cercospora beticola and powdery mildew in sugar beet.
However, the authors could further emphasize how their approach specifically advances existing methodologies or improves upon limitations in prior research. Explicitly comparing the proposed methods to similar studies would strengthen the narrative.
Rigor and Ethical Standards:

The study appears to have been conducted rigorously, and the experimental design is technically sound. The use of UAVs, NDVI indices, and multiple machine learning algorithms demonstrates a robust methodological framework.
Ethical considerations are not explicitly mentioned in the manuscript. If applicable, the authors should confirm that UAV operations and data collection complied with local regulations and ethical approval was obtained if required. Additionally, it would be beneficial to discuss any potential environmental impacts of using UAVs in agricultural settings and how these were mitigated.
Methodological Details for Replicability:

The methods section is detailed, but some areas require additional clarity to ensure full reproducibility:
The preprocessing steps for UAV spectral data should be described more comprehensively, including the software and calibration procedures used.
The criteria for selecting the spectral indices (e.g., NDVI) over others like SAVI or EVI should be elaborated upon.
The machine learning pipeline is well-described; however, further details about the training and validation process (e.g., cross-validation strategy) and hyperparameter tuning would enhance reproducibility.
Include information on how environmental variability (e.g., lighting, weather) was controlled during UAV data collection to ensure consistency.
Suggested Improvements:

Expand the introduction to highlight how this study addresses specific limitations of prior work.
Provide a more detailed description of data preprocessing and machine learning model development to ensure full reproducibility.
Include a brief statement confirming adherence to ethical and technical standards, particularly regarding UAV operations and data handling.
By addressing these points, the authors can further demonstrate the rigor and significance of their experimental design while ensuring clarity and replicability for future researcher

Validity of the findings

Impact and Novelty:

The study does not aim to assess impact or novelty explicitly, which aligns with the journal's criteria. However, the research provides a meaningful contribution to the literature by showcasing the application of UAV-based remote sensing and machine learning in the early detection of sugar beet diseases.
The findings are novel in demonstrating the integration of spectral indices and machine learning for identifying Cercospora beticola and powdery mildew. While the methods are not entirely new, the study adds value by validating their use in this specific agricultural context and encouraging replication in similar settings.
Robustness of Data and Statistical Soundness:

The data presented are robust, and the statistical analyses used to evaluate the performance of machine learning algorithms are appropriate. However, some areas need further elaboration to strengthen the findings:
Provide confidence intervals or standard errors for key results, such as classification accuracies, to enhance the statistical reliability of the findings.
Ensure all raw data are available in a discipline-specific repository with metadata for transparency and reproducibility.
Discuss potential biases or limitations in the dataset, such as environmental variability during UAV data collection or the small sample size for training machine learning models.
Conclusions and Relevance:

The conclusions are well-stated and directly linked to the research question and findings. The authors successfully demonstrate the potential of UAV-based spectral data and machine learning for agricultural disease management.
However, some conclusions extend beyond the presented data. For example, the claim that these methods can "significantly reduce economic losses" is not directly supported by quantitative evidence in the study. It would be better to rephrase this to indicate potential benefits rather than definitive outcomes.
Suggestions for Improvement:

Include a clearer discussion of the limitations of the study and how they might impact the generalizability of the findings.
Strengthen the conclusions by focusing only on results directly supported by the data. Avoid making claims that require additional evidence or broader application scenarios.
Provide detailed metadata for the raw datasets, ensuring they are robust and accessible to the research community.

Additional comments

Overall Contribution:

The manuscript addresses a relevant and timely topic in agricultural technology, specifically the application of UAV-based remote sensing and machine learning for early disease detection in sugar beet crops. This integration of technologies has significant potential for improving crop management and reducing economic losses due to diseases.
Strengths:

The study is well-structured, with a clear research question and methodology.
The use of multiple machine learning algorithms provides a comprehensive comparison of their effectiveness in disease detection.
The focus on early detection using spectral indices is practical and aligns with the needs of sustainable agriculture.
Areas for Improvement:

Language and Presentation: Minor grammatical issues and awkward phrasing should be addressed to improve readability and professionalism. A professional language review is recommended.
Figures and Tables: Improve the resolution and annotations of figures, particularly the spectral maps and performance metrics. Ensure all tables are consistently formatted and include clear explanations for all abbreviations and symbols.
Data Availability: While raw data are shared, providing detailed metadata and descriptions for the datasets would enhance reproducibility and encourage broader use of the data.
Future Directions:

The manuscript would benefit from a brief discussion of future research directions. For example, exploring the scalability of UAV-based disease detection to larger fields or applying the methods to other crops and diseases could expand the study’s impact.

Reviewer 2 ·

Basic reporting

The authors worked on sugar beets CLS and PM disease detection using MS Drone and Satellite images.
The organization of the paper is good

Experimental design

The authors used Machine Learning Algorithms for the study. Why were Machine Learning algorithms chosen over state-of-the-art Deep Learning (DL) models and algorithms?
The data preprocessing methodology requires more clarity.
How has feature correlation been performed between MS Drone images and Satellite images?

Validity of the findings

Compare the results with State of the art.

Additional comments

Conclusion should highlight the outcome of the study rather than the methods used.

Reviewer 3 ·

Basic reporting

The language requires refinement, from technical terms to overall fluency. Some sections are difficult to understand, and unnecessary details detract from the central focus of the study, which is not always clearly articulated. The introduction and especially the discussion sections lack sufficient literature to support the arguments presented. The authors have not referenced enough studies with similar objectives, and it is unclear how their findings contribute to the existing body of research. How do their results compare to those of other studies in terms of model accuracy for example? Figures need to be enhanced, with properly formatted legends and clear explanations, especially for Figures 10 and 11. The materials and methods section should include more details, while the results and discussion sections would benefit from being more concise and focused. The study’s objectives are not always clearly stated, and it is unclear whether the authors have addressed the research questions they set out to investigate.

Experimental design

The experiment is relatively straightforward, but the dataset collected has the potential to support an interesting study, given the general objectives outlined by the authors. However, the authors should present their objectives more clearly and align their analysis and results more closely with these goals. The methods section needs further clarification, particularly regarding the approach used to guide fungicide application based on UAV imagery and the training process for the disease prediction models. Additionally, the scale or level of analysis used in the study (e.g., pixel, plot, or another unit) should be explicitly stated.

Validity of the findings

The reported results, discussion, and conclusions generally align with existing literature; however, the authors have not adequately discussed how their study compares to previous research on this topic. It remains unclear how their findings contribute to the early detection of diseases in sugar beet fields, even in comparison to conventional scouting methods. The objectives should be more explicitly stated, and the findings should be directly linked to answering the research questions posed by the authors.

Additional comments

Dear Authors,

In this study, you monitored experimental plots (measuring 2.7 by 10 meters) planted with sugar beet over two growing seasons (2022 and 2023). Fungicides were applied based on two different approaches: recommendations derived from UAV (Unmanned Aerial Vehicle) multispectral images and a more conventional warning system primarily relying on climatic data. Two experiments were conducted: one focused on Cercospora leaf spot (CLS) and the other on powdery mildew (PM). Each experiment included 18 experimental plots, with six repetitions for each fungicide recommendation method, as well as a control plot where no fungicide was applied. Additionally, measurements from a handheld sensor (GreenSeeker) and data from Sentinel-2 MSI images were used for comparison with the UAV-based dataset.
Overall, the dataset appears to be complete and could serve as a relatively strong foundation for an interesting study. However, there are several methodological and conceptual issues that need to be addressed before considering this material for publication. First, the manuscript should undergo a thorough revision and proofreading process to improve language accuracy and fluidity. Additionally, the text lacks proper organization, with much of the information presented without clear division into topics. As a result, it feels cluttered, and some unnecessary details are included. The authors should focus on conveying the core message and highlighting what is essential. For instance, the term "drug" is used to refer to fungicides (page 10, line 209), and "horizontal movement" is used to describe a period of stability in crop growth (page 16, line 444). These terms are not appropriate in the given context and should be revised for clarity. Another example can be found in the Results section, where information on CLS is presented alongside mention of PM occurring in the same plots. From what I understand, you relied on natural disease occurrence rather than pathogen inoculation. However, if both diseases were present simultaneously, this could complicate or bias your analysis. It would be helpful to clarify how this issue was addressed in the study.
Another issue is the lack of clarity regarding how the UAV data was used to guide fungicide application. It appears that you employed an unsupervised classification method on NDVI images to identify areas with higher disease or stress risk. However, it seems that the plots still required visual inspection to confirm the issue. In essence, the approach appears to be more of a guided scouting method, which could be helpful but would not entirely replace the need for visual field assessments.Furthermore, you mention using Logistic Regression (LR), k-nearest neighbor classifier (KNN), Gaussian Naive Bayes (GNB), support vector machines (SVM), and decision tree classifier (DTC) to classify healthy and diseased observations. However, it is unclear how these methods were integrated into the disease monitoring process. Also, what do your observations represent? Are they individual pixels in the images or entire plots? This is quite confusing because you state that more than 19,140 "data" points were used to train these models (page 19, line 580), but the classification results you present appear to cover only a small number of observations (e.g., the confusion matrices in Table 8). Are these observations independent? Why were such a small number of observations used for validation compared to the full dataset available? Also, in the Results section you indicated that KNN provided the highest accuracy, but it seems interesting that one of the simplest approaches achieved the highest accuracy. How the models have been parametrized? Have you used grid search for example to optimize model parameters in each case? Additionally, it would be helpful to know what software or environment was used to conduct this part of the analysis. Clarifying these aspects will help improve the transparency and reproducibility of the methodology.
Figure 10 is also quite intriguing. From what I understand, you present a time-series of average NDVI values for each ‘treatment’ (i.e., the method used to determine the need for fungicide application). However, it is striking that there are large differences between the various sensing solutions. Could you provide an explanation for these differences? Additionally, I am curious about how you managed to extract data from Sentinel-2 images for your plots, given that the plots are smaller than the sensor's pixel size, as far as I understand. Could you clarify how this issue was addressed? Lastly, the manuscript seems to place significant emphasis on analyzing the ‘treatment’ level, but it might be more important to focus on disease severity and occurrence. A more direct link between NDVI values and disease progression could provide a clearer picture of the efficacy of the fungicide application methods. Furthermore, is NDVI sufficient for early disease detection? It’s important to be cautious here, as early detection in the literature generally refers to pre-visual symptoms. Could you discuss whether NDVI values can reliably identify diseases before visible symptoms appear?
Figure 11 is quite difficult to interpret. You use it to suggest that “…temperature and humidity are important parameters in determining the disease (Figure 11)” (page 20, line 595). However, it’s unclear what the heatmaps in this figure actually represent. Do they show the correlation between the variables used in the models? Are these the only variables that are important for the predictions? This aspect should be more thoroughly explained to ensure clarity for the reader. A clearer description of what the heatmaps represent and how they relate to the disease prediction would greatly improve the interpretation of this Figure.
I will not provide more specific comments on the text at this stage, as I believe the material should be reformulated in line with the points mentioned above. A more thorough revision is needed before a careful evaluation can be made.

Kind Regards,

---

## Round 0.2 · Minor Revisions

I appreciate your constructive attitude toward the reviewers' suggestions and your improvement of your article based on them. After a few corrections, I believe your manuscript will be ready for publication.

Our section Editor Robert Winkler highlighted some points in your article such as below:

> The research presented in this manuscript is both well-conducted and substantially significant, representing a noteworthy contribution to the scientific community. However, to fully highlight the importance and impact of the work, it is recommended that the writing style be refined for greater concision and clarity. For instance, instead of using phrases like "The findings of this research show that...", consider a more direct approach such as "UAV-based images show that...". By streamlining sentences and eliminating filler phrases such as "It has been confirmed that" and "The research provides valuable insights," the tone of the paper can be made more assured and the manuscript's readability improved.
>
> Authors should also spell out abbreviations and acronyms at first use - for instance, 'Uncrewed Aerial Vehicles" in the title.

I recommend you read these suggestions and improve your article based on them. If you find yourself in disagreement with any particular suggestion, it would be beneficial to provide clear and well-reasoned justifications for your perspective.

·

Basic reporting

thank you very much to the Authors, They are made all the comments as we suggested.and I have no comments

Experimental design

No comments also they made all modifications

Validity of the findings

I have nothing to add the Author made all the suggestion

---

## Round 0.3 · Minor Revisions

I would like to thank you for accepting the referees' suggestions and improving your article based on their suggestions. Our section editor and I believe your article requires vigorous proofreading and editing to improve the writing style. I recommend seeking assistance from a colleague or utilizing an editing service to ensure the language is professional.

**Language Note:** The Academic Editor has identified that the English language must be improved. PeerJ can provide language editing services - please contact us at [email protected] for pricing (be sure to provide your manuscript number and title). Alternatively, you should make your own arrangements to improve the language quality and provide details in your response letter. – PeerJ Staff

---

## Round 0.4 · accepted · Accept

I would like to thank you for accepting the referees' suggestions and improving your article based on their suggestions. Your article is ready to publish. We look forward to your next article.